# OpenReview forum: "Locks Tested Without Burglars: Using Coding Assistants to Break Prompt Injection Defenses"
_NeurIPS.cc/2025/Workshop/Reliable_ML — NeurIPS 2025 - Reliable ML Workshop_

### Official Review · Reviewer_FoTW · 2025-09-12
**Strong paper that highlights weakness in existing evaluation of prompt injection defenses.**

**Rating:** 7
**Confidence:** 3

**Review:**

1. This paper questions the evaluation methodology used by work proposing defenses against prompt injection tasks. While most evaluations of prompt injection defenses are done on static toy benchmarks, this paper shows that one can easily bypass these proposed defenses by using AI coding assistants as automated red-teamers. The main takeaway is that defenses against prompt injection attacks should be evaluated against adaptive attacks such as those by coding-assistant-driven red-teaming.

2.  I think this paper is an important contribution to the prompt injection research community. As far as I can tell (I am not an expert in this area), the idea of using a Coding agent as a red-teamer is novel, and the results are strong. I think this paper brings up an important and realistic safety concern that should be addressed when evaluating prompt-injection defenses.

3. Missing comparisons. I think this paper would benefit from a more in-depth related works section, as some claims made by the authors are not fully substantiated.

4. As someone who is not an expert in the field, I think it would be good if the authors can further substantiate their repeated claims that "in prompt injection research, such adaptive evaluation is rare; most defenses are only tested against a toy benchmark." A more in-depth related work section is needed to convince the reader that this statement is indeed true. Likewise, it would be helpful if the authors can expand more about existing adaptive attack strategies (surely there are some?)

---

### Official Review · Reviewer_Sawp · 2025-09-22
**interesting results**

**Rating:** 6
**Confidence:** 2

**Review:**

1. This work criticizes existing evaluation practice for "adversarial defenses" and shows that current coding assistants can effectively construct tailored attacks to those defenses.
2. The results themselves are interesting and important. The paper is overall easy to follow.
3. I understand adversarial robustness is listed in the CFP, but I personally am not entirely sure whether this paper addresses the problem of "unreliable data".

---

### Official Review · Reviewer_zxv4 · 2025-09-24

**Rating:** 7
**Confidence:** 4

**Review:**

The paper explores the idea that AI coding assistants (e.g., Claude Code) can act as automated red-teamers for prompt injection defenses. Unlike toy benchmarks, coding assistants can parse defense codebases, extract hidden prompts/assumptions, and generate adaptive, natural-language attacks. The authors evaluate three state-of-the-art defenses (DataSentinel, MELON, DRIFT) across both their original benchmarks and more realistic settings (AgentDojo, CrAIBench). Results show that assistant-generated adaptive attacks dramatically raise attack success rates. It argues for a new evaluation paradigm where defenses must be tested against automated red-teaming before robustness claims are made.

Strengths:
1. Frames coding assistants as practical adversarial collaborators, not just productivity tools, providing a novel lens on LLM security.
2. Demonstrates large gaps between benchmark-based evaluation and adaptive adversarial evaluation.
3. Includes both toy and realistic tasks.

Weaknesses:
1. Only Claude is tested. A larger range of models could be tested.
2. Only three defense techniques are tested.